# Beyond Classification Accuracy: Neural-MedBench and the Need for Deeper Reasoning Benchmarks

**Miao Jing**[†,1,5]**, Mengting Jia**[†,2]**, Junling Lin**[3]**, Zhongxia Shen**[4]**, Huan Gao**[2,6]**,
Mingkun Xu**[2,*]**, Shangyang Li**[1,2,7,*]

[1]School of Physics Science and Technology,
Beijing University of Posts and Telecommunications, Beijing, China
[2]Guangdong Institute of Intelligence Science and Technology, Hengqin, Zhuhai, China
[3]Beijing Chaoyang Hospital, Capital Medical University, Beijing, China
[4]Sleep Medical Center, Huzhou Third Municipal Hospital,
Affiliated Hospital of Wenzhou Medical University, Huzhou, China
[5]University of Macau, Macau, China
[6]Renyixun Health Technology Co., Ltd, Beijing, China
[7]Academy for Advanced Interdisciplinary Studies, Peking University, Beijing, China
[†]Equal contribution    [*]Corresponding authors
shangyang_li@foxmail.com, xumingkun@gdiist.cn

## Abstract

Recent advances in vision-language models (VLMs) have achieved remarkable performance on standard medical benchmarks, yet their true clinical reasoning ability remains unclear. Existing datasets predominantly emphasize classification accuracy, creating an evaluation illusion in which models appear proficient while still failing at high-stakes diagnostic reasoning. We introduce Neural-MedBench, a compact yet reasoning-intensive benchmark specifically designed to probe the limits of multimodal clinical reasoning in neurology. Neural-MedBench integrates multi-sequence MRI scans, structured electronic health records, and clinical notes, and encompasses three core task families: differential diagnosis, lesion recognition, and rationale generation. To ensure reliable evaluation, we develop a hybrid scoring pipeline that combines LLM-based graders, clinician validation, and semantic similarity metrics. Through systematic evaluation of state-of-the-art VLMs, including GPT-4o, Claude-4, and MedGemma, we observe a sharp performance drop compared to conventional datasets. Error analysis shows that reasoning failures, rather than perceptual errors, dominate model shortcomings. Our findings highlight the necessity of a Two-Axis Evaluation Framework: breadth-oriented large datasets for statistical generalization, and depth-oriented, compact benchmarks such as Neural-MedBench for reasoning fidelity. We release Neural-MedBench at https://neuromedbench.github.io/ as an open and extensible diagnostic testbed, which guides the expansion of future benchmarks and enables rigorous yet cost-effective assessment of clinically trustworthy AI.

## 1 Introduction

Recent advances in vision-language models (VLMs) have led to striking improvements across a wide range of medical AI tasks. On standard benchmarks such as MedMNIST v2 Yang et al. (2023) and MultiMedQA Singhal et al. (2023), state-of-the-art models achieve near-human or even superhuman performance in label prediction and image–text alignment. These results have created an impression that medical VLMs are nearing clinical readiness. Yet, as the clinical reasoning literature has recently underscored Schwartzstein (2024), safe and effective diagnostic practice, especially in high-stakes fields such as neurology, demands more than classification accuracy: it requires multimodal synthesis, ambiguity resolution, and the capacity to justify conclusions in a manner consistent with clinical logic. Current benchmarks, despite their scale, rarely capture these aspects. We argue that this discrepancy

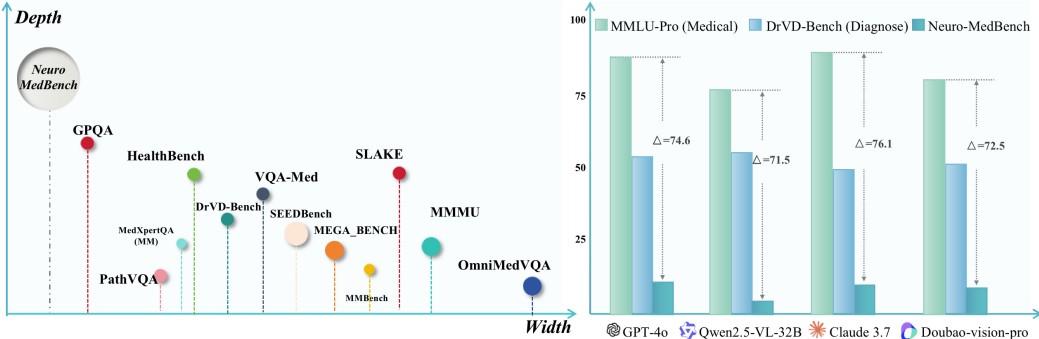

Figure 1: **Conceptual Positioning of `Neural-MedBench` and Empirical Evidence of the Evaluation Illusion.** (Left) An illustrative mapping of existing medical VLM benchmarks along the proposed axes of Breadth (dataset size and diversity) and Depth (reasoning complexity). Most benchmarks occupy the high-Breadth, low-Depth space, leaving a critical gap in high-Depth evaluation, which `Neural-MedBench` is designed to fill. (Right) Performance of leading VLMs on two Breadth-Axis benchmarks (MMLU-Pro and DrVD-Bench) versus our Depth-Axis benchmark (`Neural-MedBench`). The stark performance drop on Neural-MedBench provides clear empirical evidence of the "evaluation illusion" and the disconnect between the two evaluation axes. Table.2 shows full results.

creates an *evaluation illusion*, a misleading sense of model capability, where strong performance on shallow benchmarks obscures deep weaknesses in reasoning fidelity and clinical trustworthiness.

To address this gap, we propose a new perspective on evaluation: a **Two-Axis Evaluation Framework** for medical AI. Along the first axis, **Breadth**, large-scale datasets assess statistical generalization and coverage across populations and conditions. This is where almost all existing benchmarks operate. However, recent work such as DiagnosisArena Zhu et al. (2025) has shown that relying solely on such breadth-oriented evaluations leads to overestimation of model capability, as they fail to capture true clinical reasoning challenges. Along the second axis, **Depth**, compact but diagnostically complex, expert-curated benchmarks assess reasoning fidelity, forcing models to integrate multimodal signals, reason under uncertainty, and provide structured justifications, an approach that echoes trends in MedAgentsBench Tang et al. (2025). Importantly, we hypothesize that these axes are largely uncorrelated: success on breadth benchmarks does not guarantee competence in depth scenarios. This hypothesis is supported by findings that even expert LLMs struggle with self-awareness and metacognitive reasoning on challenging cases, as shown in MetaMedQA Griot et al. (2025). A complete picture of model readiness therefore requires evaluation along both dimensions.

To operationalize the Depth axis in clinical neurology, we introduce `Neural-MedBench`, a diagnostic benchmark deliberately designed as a "stress test". `Neural-MedBench` comprises 120 expert-annotated multimodal cases, including multi-sequence MRI scans, structured electronic health records (EHRs), and clinical narratives, yielding 200 reasoning-intensive tasks. Its design mirrors practices in medical education, such as Objective Structured Clinical Examinations (OSCEs), where carefully constructed cases are used to assess reasoning skills under uncertainty Geathers et al. (2025); Siegelman et al. (2024). Unlike prior resources focused on breadth, `Neural-MedBench` is compact by design, prioritizing reasoning density over volume, enabling cost-effective yet high-signal evaluation of clinical reasoning.

Our evaluation of leading models, including GPT-4o, Claude 4, and MedGemma, reveals a striking pattern: models that excel on large-scale benchmarks fail systematically on `Neural-MedBench`. Error analysis shows that these failures stem not from perception or lexical mismatch, but rather from breakdowns in clinical reasoning, limitations that remain hidden under existing evaluation paradigms. This disconnect provides the first empirical evidence for the independence of the two axes, underscoring the urgent need for benchmarks that explicitly capture reasoning depth.

This work makes the following contributions:

- We propose the Two-Axis Evaluation Framework, arguing that trustworthy clinical AI requires complementary assessments of both breadth (statistical generalization) and depth (reasoning fidelity).

- We release `Neural-MedBench`, the first neurology-focused benchmark explicitly designed to operationalize the Depth axis, comprising 120 multimodal, expert-curated diagnostic cases with 200 reasoning-intensive tasks.

- We provide empirical evidence of the disconnect between breadth and depth, showing that state-of-the-art VLMs fail primarily at reasoning, despite strong performance on existing large-scale datasets.

- We present a systematic error analysis and a human performance baseline to contextualize model results, and we release `Neural-MedBench` as an open, extensible resource with a public leaderboard and roadmap for expansion.

## 2    RELATED WORK

### 2.1    MEDICAL VLM BENCHMARKS

A wide range of benchmarks have been developed for evaluating medical VLMs. Early datasets such as ChestX-ray14 Wang et al. (2017), CheXpert Irvin et al. (2019), and MIMIC-CXR Johnson et al. (2019) primarily enabled large-scale classification and report generation. Organ- or disease-specific challenges—including LUNA for pulmonary nodules Setio et al. (2017), LiTS for liver tumors Bilic et al. (2019), BraTS for brain tumors Menze et al. (2015), and ISLES for stroke Maier et al. (2017)—expanded evaluation to segmentation and lesion characterization. More recent multimodal benchmarks target richer tasks: ROCO v2 Rückert et al. (2024) for radiology captioning, RadGraph Jain et al. (2021) for entity–relation extraction, and MedFMC Wang et al. (2023) for federated multi-center classification. VQA-style benchmarks, such as VQA-RAD Lau et al. (2018), PathVQA He et al. (2020), and OmniMedVQA Hu et al. (2024), test text–image reasoning but often rely on templated questions, limiting their coverage of authentic diagnostic challenges. Recent meta-benchmarks (e.g., Rad-Bench Kuo et al. (2024), DiagnosisArena Zhu et al. (2025), MedAgents-Bench Tang et al. (2025)) have emphasized comprehensiveness, yet remain breadth-oriented: they focus on scale and diversity rather than probing reasoning under uncertainty.

In contrast, `Neural-MedBench` explicitly operationalizes the *Depth axis* of evaluation. By curating 120 multimodal, diagnostically complex neurology cases into 200 reasoning-intensive tasks, it emphasizes reasoning fidelity over volume. This design mirrors clinical examinations such as OSCEs, and captures underexplored challenges like multi-lesion differential diagnosis, contextual interpretation, and longitudinal integration across MRI sequences.

### 2.2    VLM EVALUATION METHODOLOGIES

Medical VLMs are typically evaluated via three strategies: reference-based, reference-free, or hybrid metrics. Traditional reference-based methods include BLEU, ROUGE, and CIDEr for surface overlap, as well as semantic similarity scores such as BERTScore Zhang et al. (2019), CLIPScore Hessel et al. (2021), and domain-specific adaptations like CheXbert Smit et al. (2020) or RadGraph-F1 Jain et al. (2021). While scalable, these methods cannot assure that generated outputs are clinically grounded. Reference-free approaches, such as saliency attribution (e.g., Grad-CAM Selvaraju et al. (2020)) or image–text alignment tests, serve as indirect proxies for reasoning but lack explanatory power. More recent frameworks have introduced LLM-based grading systems for evaluating factual accuracy and reasoning plausibility in free-form clinical responses. For example, the Expert-of-Experts Verification and Alignment (EVAL) framework streamlines clinician-led evaluation in bleeding scenarios Giuffrè et al. (2025), while LLMEval-Med combines expert-designed checklists with LLM-as-Judge dynamics to improve reliability in medical QA and reasoning tasks Zhang et al. (2025); Zhi et al. (2025); Fang et al. (2026); Zheng et al. (2026). Despite their automation advantages, these approaches remain subject to ongoing reliability validation.

`Neural-MedBench` employs a hybrid evaluation strategy: semantic fidelity is quantified using BERTScore, while clinical reasoning is scored via an LLM-based evaluator calibrated with neurologist

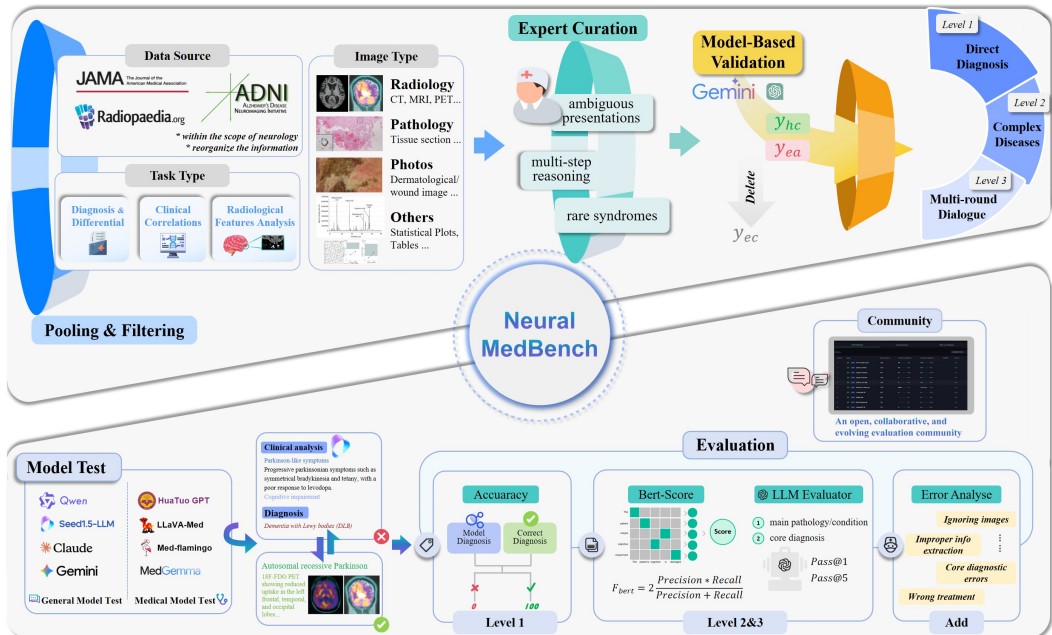

Figure 2: **Overview of `Neural-MedBench`.** The workflow begins with pooling and filtering data from diverse clinical sources, followed by a multi-stage expert curation and model-based validation pipeline to select for diagnostically complex cases. The resulting benchmark is used to test a cohort of models, whose outputs are assessed via a hybrid evaluation suite including accuracy, semantic similarity, and a clinician-validated LLM grader.

review. This design allows scalable yet clinically faithful assessment of both "what" is predicted and "why," exposing nuanced failure modes that conventional metrics tend to overlook.

## 3 Neural-MedBench

**Design Philosophy and Motivation.** As established in our proposal for a Two-Axis Evaluation Framework, current medical benchmarks overwhelmingly emphasize Breadth, rewarding statistical generalization but failing to probe deep, integrative reasoning. To address this gap, `Neural-MedBench` was conceived to operationalize the Depth axis. Its design philosophy is therefore fundamentally different from large-scale datasets.

Instead of prioritizing data volume, we prioritize reasoning density. Our goal was to create a compact but diagnostically rich "stress test," inspired by high-stakes clinical examinations like OSCEs, where the ability to synthesize, reason, and justify is paramount. This approach, illustrated in Figure 2, allows for a rigorous yet cost-efficient evaluation of the core clinical reasoning capabilities that remain under-assessed by existing resources.

**Case Sources and Curation.** We curated 120 authentic neurology cases from multiple rigorously vetted sources: ADNI and OASIS (research cohorts), Radiopaedia (expert-verified imaging cases), and peer-reviewed case reports (e.g., JAMA Neurology). These sources span both common diseases (e.g., Alzheimer's disease, ischemic stroke, epilepsy) and rare or diagnostically complex conditions (e.g., autoimmune encephalitis, CNS infections).

The final 120 cases were selected from an initial pool of over 2,000 candidates using a funnel-shaped, multi-stage pipeline: (1) *Initial screening* retained only cases with sufficient multimodal completeness (e.g., imaging, neuropsychological scores, patient histories); (2) *Expert curation* by two senior neurologists and one neuroradiologist, who reviewed cases for plausibility, diagnostic complexity, and educational value; (3) *Annotation* of ground-truth answers, including final diagnosis, differential diagnoses, lesion characterization, and explanatory reasoning; (4) *Consensus review and*

*challenge validation*, where disagreements were resolved through discussion, and trivial cases filtered out using baseline models, ensuring each retained case posed a meaningful diagnostic challenge.

This rigorous process ensures that ground truths are not merely single labels but structured narratives that model how neurologists articulate reasoning. Each case supports on average 2 reasoning tasks (200 tasks in total), spanning direct diagnosis, complex disease inference, and multi-turn dialogue. This compact yet high-density design prioritizes diagnostic richness and evaluation efficiency, enabling fine-grained reasoning assessment without the prohibitive annotation and compute costs of large-scale datasets.

**Task Families and Difficulty Stratification.** From the curated cases, we derived 200 reasoning-intensive tasks, grouped into three core families:

| Task Family | Description |
|---|---|
| *Differential Diagnosis* | Given imaging and patient history, models must provide a ranked diagnostic hypothesis with justification. |
| *Lesion Recognition* | Identify lesion type and location, testing multimodal spatial reasoning. |
| *Rationale Generation* | Generate explanatory reasoning for diagnostic choices, mirroring case discussions or board exams. |

To probe model reasoning at different depths, these tasks are further stratified into three difficulty levels (see examples in Fig. 3):

- **Level 1 (Direct Diagnosis):** Involves straightforward cases with classic, unambiguous signs primarily in a modality (e.g., a clear tumor on an MRI). This level tests core pattern recognition.

- **Level 2 (Complex Diagnosis):** Features cases with ambiguous or conflicting evidence that require integrating information from at least two modalities (e.g., linking subtle imaging findings with specific details from the clinical history) to resolve the uncertainty.

- **Level 3 (Iterative Diagnosis):** Simulates a multi-turn clinical consultation where the model must progressively refine its diagnosis by interpreting new information. This tests its ability to dynamically adjust its reasoning chain.

This design provides a controlled lens into how models scale from factual recall to higher-order clinical reasoning.

**Evaluation Protocol.** Evaluating deep clinical reasoning is notoriously resource-intensive. To address this, `Neural-MedBench` introduces a scalable, hybrid evaluation pipeline that is rigorously validated by clinicians, but fully automatable for community use. The protocol consists of two stages:

*Stage 1: Grader Validation.* The core of our protocol is a dedicated LLM-based grader, guided by detailed, neurology-specific rubrics. To ensure this automated grader serves as a reliable proxy for expert judgment, we performed a rigorous clinician-in-the-loop validation. A panel of board-certified neurologists independently scored a large subset of model outputs using the same rubric. We found a very high correlation between the LLM grader's scores and the consensus of human experts (e.g., Pearson's $r > 0.9$), scientifically validating our grader as a robust instrument for automated assessment.

*Stage 2: Automated Community Evaluation.* Following this one-time, intensive validation, the clinically-calibrated LLM grader is released as part of the benchmark. This allows any researcher to evaluate a new model in a fully automated fashion, without requiring access to clinicians. Users can simply run their model on `Neural-MedBench` and use our provided, validated grader to obtain scores that are highly correlated with expert neurological assessment.

To further contextualize results from this automated pipeline, we also establish a human performance baseline by evaluating clinicians of varying expertise on the benchmark. This crucial addition anchors all model scores, providing a clear, realistic measure of the gap between current AI capabilities and human-level clinical reasoning. This two-stage protocol thus offers the best of both worlds: the gold-standard rigor of clinician validation and the scalability of automated evaluation.

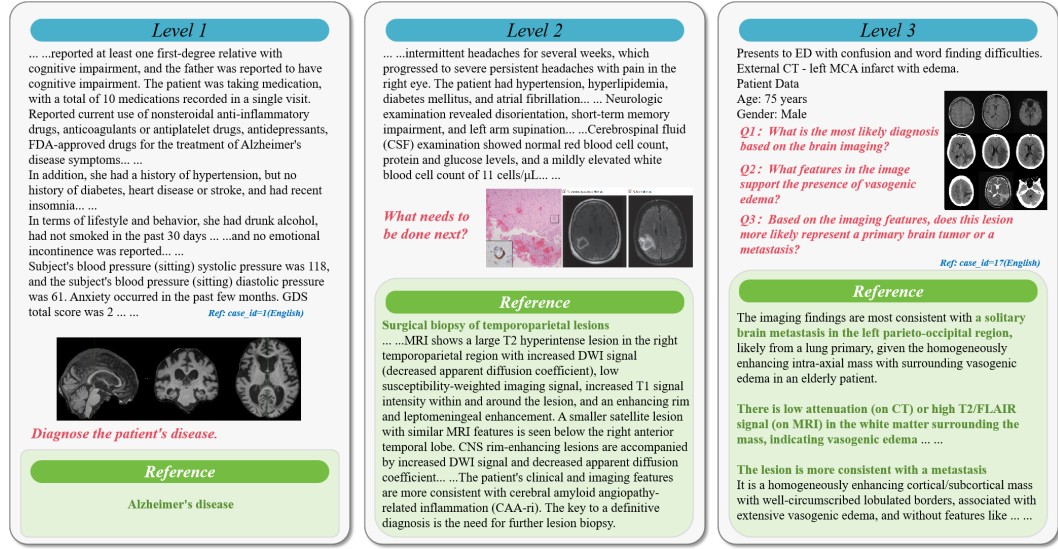

Figure 3: **Examples of questions and their reference answers in `Neural-MedBench`, illustrating typical reasoning tasks.**

**Dataset Analysis.** Table 1 summarizes task statistics. Text inputs were normalized in length to control for spurious effects of input size on model performance. Imaging data include balanced representation of T1, T2, FLAIR, and CT modalities. Each difficulty tier contains both common and rare pathologies, ensuring robustness tests beyond frequency bias.

Table 1: Text Lengths and Image Pixels by Subset

| Subset | Token Chinese | Token English | Image Pixel (x10,000) |
|--------|---------------|---------------|------------------------|
| Level 1 | 374 | 225 | 535 |
| Level 2 | 504 | 309 | 145 |
| Level 3 | 67 | 37 | 1147 |

**Summary.** By integrating multimodal data, stratified diagnostic difficulty, structured reasoning targets, and a cost-efficient hybrid evaluation protocol, `Neural-MedBench` operationalizes the Depth axis of clinical AI evaluation. It complements large-scale breadth benchmarks with a compact, reproducible, and extensible testbed for probing reasoning fidelity, advancing toward clinically trustworthy multimodal AI.

## 4 EXPERIMENTAL SETUP

Our experimental setup is designed to comprehensively assess the clinical reasoning capabilities of state-of-the-art VLMs on `Neural-MedBench`, and to contextualize their performance against a realistic human baseline.

### 4.1 MODELS BENCHMARKED

We evaluate 16 representative VLMs spanning three categories, all in a zero-shot setting to probe intrinsic reasoning capabilities without task-specific fine-tuning:

We benchmark three categories of models to provide a comprehensive view of current VLM capabilities. First, we include proprietary frontier systems with strong multimodal reasoning ability, namely **GPT-Series ?** and **Gemini** Comanici et al. (2025), which serve as reference baselines. Second, we evaluate powerful open-source generalist models such as **Claude** Anthropic (2024), **Doubao** Team (2024), **Qwen-VL-Plus** Bai et al. (2023). Finally, to assess the effect of domain-specific adaptation,

we benchmark medical-specialized VLMs including **LLaVA-Med** Li et al. (2023), **RadFM** Wu et al. (2023), **Huatuo-GPT-Vision** Chen et al. (2024), **Med-Flamingo** Moor et al. (2023), **Lingshu** LASA-Team et al. (2025) and **MedGemma** Sellergren et al. (2025), which represent current efforts to tailor VLMs for clinical applications. A full list of models, versions, and hyperparameters is provided in Appendix C.

## 4.2 HUMAN PERFORMANCE BASELINE

To establish a realistic anchor for task difficulty, we conducted a human evaluation with two clinician groups: Medical Students ($n = 5$) and Senior Physicians ($n = 5$). Each participant completed the entire set of 200 tasks, blinded to both model outputs and gold-standard answers. All responses were scored using the same evaluation protocol as applied to the models, ensuring a direct and fair human–model comparison.

## 4.3 TASK AND PROMPTING STRATEGY

To simulate clinical consultations, we adopt a structured role-prompting approach. Each model is initialized as an "experienced neurologist" to encourage adherence to medical reasoning and terminology. Tasks are presented in full, combining textual narratives, structured patient history, and imaging (MRI/CT encoded in base64). Full prompt templates are detailed in Appendix B.

## 4.4 EVALUATION METRICS

Given the multi-faceted nature of clinical reasoning, we employ a suite of complementary metrics:

- Diagnostic Accuracy (pass@k): For tasks with definitive outcomes, we report top-1 (pass@1) and top-5 (pass@5) accuracy. The latter measures whether the correct diagnosis appears within a model's differential.
- Semantic Fidelity (BERTScore): Used for free-form rationale generation, capturing semantic alignment between model outputs and expert references. We use it only as a secondary similarity indicator.
- Reasoning Fidelity (LLM Grader): A clinically calibrated GPT-4o grader evaluates correctness, logical coherence, and evidence grounding (see Section 5.2).
- Error Taxonomy: To understand failure modes, incorrect responses were manually annotated by neurologists into five categories: *Perceptual Failure*, *Reasoning Failure*, *Knowledge Gap*, *Grounding Error*, and *Visual Hallucination*.

This combination of automated metrics, human baselines, and error analysis provides a rigorous and clinically grounded evaluation protocol.

# 5 EXPERIMENTAL RESULTS AND ANALYSIS

Our experimental analysis is structured to answer three central questions: (1) How do state-of-the-art VLMs perform on a Depth-Axis benchmark like `Neural-MedBench`, and does this performance align with their success on Breadth-Axis benchmarks? (2) How large is the performance gap between these models and practicing human clinicians? (3) What are the primary underlying reasons for model failures in complex clinical reasoning?

## 5.1 OVERALL PERFORMANCE: A STARK DISCONNECT BETWEEN BREADTH AND DEPTH

We evaluated a diverse cohort of VLMs on all 200 tasks in `Neural-MedBench`, with full results in Table 2. The findings reveal a profound difficulty that stands in stark contrast to conventional benchmarks. Across all models, pass@1 accuracy remains strikingly low, with even the top-performing specialized model MedGemma-27B-it only reaching 30.0% on the simplest *Direct Diagnosis* tasks. Performance plummets on more complex subsets, averaging below 15% for most models, a stark contrast to the Senior Physician's performance of over 35% on the same tasks. Allowing for five attempts (*pass@5*) improves scores, with the generalist model Gemini 2.5-Pro achieving the highest

score of 50%. However, no model consistently surpasses the 50% mark, indicating that the correct diagnosis is often not even within their top considerations.

Table 2: Evaluation metrics across different models and tasks

| Models | Direct diagnosis | | Complex diseases | | | Multi-round dialogue | | |
|---|---|---|---|---|---|---|---|---|
| | pass@1[%](n) | pass@5[%](n) | BertScore | pass@1[%](n) | pass@5[%](n) | BertScore | pass@1[%](n) | pass@5[%](n) |
| **Base VLMs** | | | | | | | | |
| GPT-5 | **36.7 (11)** | 43.3 (13) | **0.80** | **28.3 (17)** | **45.0 (27)** | **0.81** | **19.5 (39)** | **27.5 (55)** |
| GPT-4o | 20.0 (6) | 36.7 (11) | 0.70 | 8.3 (5) | 40.0 (24) | 0.73 | 8.5 (17) | 16.5 (33) |
| Gemini 2.5-Pro | 30.0 (9) | **50.0 (15)** | 0.77 | 15.0 (9) | 38.3 (23) | 0.72 | 11.5 (23) | 19.5 (39) |
| Gemini 2.5-Flash | 26.7 (8) | 46.7 (14) | 0.76 | 13.3 (8) | 35.0 (21) | 0.68 | 10.5 (21) | 18.5 (37) |
| **General VLMs** | | | | | | | | |
| Gemini 2.0-Flash | 20.0 (6) | 30.0 (9) | 0.68 | 11.7 (7) | 23.3 (14) | 0.67 | 9.0 (18) | 16.0 (32) |
| Claude 4.0-Sonnet | 16.7 (5) | 43.3 (13) | 0.78 | 13.3 (8) | 31.6 (19) | 0.77 | 6.5 (13) | 18.0 (36) |
| Claude 3.7 Sonnet | 16.7 (5) | 26.7 (8) | 0.73 | 8.3 (5) | 31.7 (19) | 0.66 | 7.5 (15) | 12.0 (24) |
| Claude 3.5 Sonnet | 6.7 (2) | 16.7 (5) | 0.68 | 8.3 (5) | 18.3 (11) | 0.66 | 7.0 (14) | 10.5 (21) |
| Qwen-VL-2.5 | 10.0 (3) | 30.0 (9) | 0.79 | 5.0 (3) | 21.7 (13) | 0.69 | 4.0 (8) | 10.0 (20) |
| Doubao-1.5-vision-pro | 6.7 (2) | 40.0 (12) | 0.64 | 10.0 (6) | 13.3 (8) | 0.63 | 5.5 (11) | 11.5 (23) |
| **Medical VLMs** | | | | | | | | |
| LLaVA-Med | 10.0 (3) | 16.7 (5) | 0.66 | 10.0 (6) | 28.3 (17) | 0.69 | 6.0 (12) | 13.0 (26) |
| MedGemma | 30.0 (9) | 36.7 (11) | **0.80** | 18.3 (11) | 38.3 (23) | 0.75 | 10.5 (21) | 15.5 (31) |
| Lingshu | 26.7 (8) | 40.0 (12) | 0.75 | 21.7 (13) | 35.0 (21) | 0.73 | 8.5 (17) | 20.0 (40) |
| RadFM | 0.0 (0) | 20.0 (6) | 0.62 | 3.3 (2) | 13.3 (8) | 0.67 | 2.5 (5) | 6.0 (12) |
| Med-Flamingo | 0.0 (0) | 16.7 (5) | 0.67 | 3.3 (2) | 10.0 (6) | 0.66 | 1.5 (3) | 10.0 (20) |
| HuatuoGPT | 10.0 (3) | 20.0 (6) | 0.60 | 5.0 (3) | 13.3 (8) | 0.68 | 3.0 (6) | 7.0 (14) |
| **Human** | | | | | | | | |
| Medical Student | 3.3 (1) | —— | —— | 3.3 (2) | —— | —— | 6.0 (12) | —— |
| Senior Physician | 40.0 (12) | —— | —— | 35.5 (21) | —— | —— | 15.0 (30) | —— |

This stark performance drop provides the first empirical evidence for our central hypothesis: the two evaluation axes (Breadth and Depth) are largely uncorrelated. To visualize this "evaluation illusion," Figure 1 contrasts the high scores on a typical Breadth-Axis benchmark against the low scores on `Neural-MedBench`, demonstrating that success in pattern recognition does not translate to competence in deep clinical reasoning.

## 5.2 THE GAP TO HUMAN EXPERTISE: A SOBERING REALITY

To provide a crucial, real-world anchor for model performance, we benchmarked leading VLMs against both the ideal Ground Truth (expert consensus) and realistic human baselines. The results, visualized in Figure 4(Left), reveal a sobering chasm. On Direct Diagnosis tasks, the top-performing VLM, MedGemma, reached 30.0% pass@1 accuracy, falling 10 percentage points short of the Senior Physician's 40.0% and barely surpassing a Medical Student. This performance gap widens dramatically in the *Complex Diseases* subset, where the Senior Physician's performance (35.5%) is nearly double that of MedGemma (18.3%). This quantitative and clinically meaningful gap validates our benchmark's difficulty and underscores the immense distance to human-level clinical competence.

The performance gap between medical students and VLMs reflects distinct reasoning paradigms. Students' lower $pass@1$ scores stem from clinical training that prioritizes broad differentials to avoid premature closure, whereas VLMs benefit from exam-style pattern matching for decisive answers. However, students are better in multi-turn dialogues due to superior metacognition—the ability to self-correct using new evidence. In contrast, VLMs exhibit significant anchoring bias, often failing to revise initial hypotheses even when presented with disconfirming clinical data.

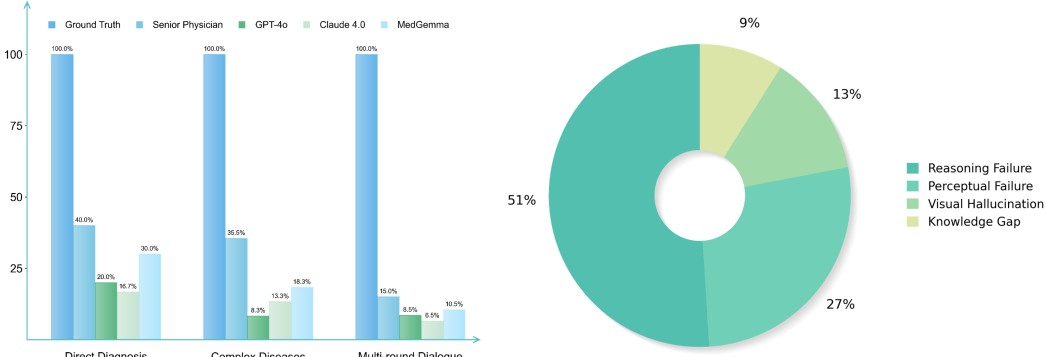

Figure 4: **The Performance Gap to Human Expertise and a Systematic Diagnosis of Model Failures.** (Left) A direct comparison of pass@1 accuracy. The "Ground Truth" represents the expert consensus answer. The "Senior Physician" bar shows the performance of a board-certified neurologist in a blinded test, providing a realistic human expert baseline. All evaluated VLMs perform significantly below this realistic baseline, widening the gap as task complexity increases. (Right) Distribution of primary error types from a systematic analysis of incorrect model responses. Reasoning Failure is the dominant cause of error, suggesting the primary bottleneck is cognitive, not perceptual.

## 5.3 WHY DO MODELS FAIL? A SYSTEMATIC ERROR ANALYSIS

To diagnose the root cause of this performance gap, we performed a systematic analysis of 100 incorrect model responses. The distribution of primary error types (Figure 4(Right)) reveals that the primary bottleneck for current VLMs is not perception, but reasoning. A striking 51% of failures were classified as Reasoning Failures, where models correctly identified key findings but failed to synthesize them into a correct diagnosis. This rate is nearly double that of Perceptual Failures (27%). This key finding allows us to define the "lower bound" of current VLM capabilities. Our results suggest this capability floor is set not by a lack of perceptual acuity, but by a fundamental deficit in integrative clinical reasoning. This directly explains the observed disconnect between high linguistic fluency and low diagnostic accuracy, highlighting the necessity of Depth-Axis benchmarks for diagnosing and ultimately fixing these core reasoning deficiencies.

## 5.4 ADDITIONAL FINDINGS

**Foundation Models vs. Medical-Specialized Models.** Our results reveal a nuanced relationship between generalist and specialist VLMs. On single-best-guess accuracy (pass@1), the medical-specialized MedGemma-27B-it is good, demonstrating the value of domain-specific fine-tuning for diagnostic precision. However, on differential diagnosis generation (pass@5), large generalist models like Gemini 2.5-Pro retain an edge, likely due to their broader generative capabilities. This suggests that optimal clinical AI may require a synthesis of both deep specialization and generative breadth.

**Evaluation Efficiency.** As a Depth-Axis benchmark, `Neural-MedBench` is deliberately designed for high signal-to-noise ratio and evaluation efficiency. Our diagnostic prompts are carefully engineered to elicit rich, multi-faceted reasoning from a compact set of cases. As summarized in Table 3, this high-density design reduces inference token costs by over an order of magnitude (10x) compared to large-scale Breadth-Axis benchmarks like GMAI-MMBench Ye et al. (2024) and Omni-iMedVQA Hu et al. (2024), while maintaining high diagnostic difficulty. This cost-effectiveness is a key feature, as it lowers the barrier for academic labs and enables practical, multi-sample robustness analyses (e.g., via temperature sweeps), which are often computationally prohibitive on larger datasets but are essential for assessing model reliability.

Table 3: GPT-4o performance on different benchmarks

| Benchmark | Number of Images | Cost of Image Tokens | Passing Rate (%) |
|---|---|---|---|
| GMAI-MMBench | 12K | $30.00 | 53.96 |
| OmniMedVQA | 128K | $320.00 | 29.74 |
| Neural-MedBench | 1K | $2.50 | 9.67 |

## 6  LIMITATIONS AND FUTURE WORK

While `Neural-MedBench` was deliberately curated for diagnostic depth, we acknowledge several limitations in its current composition. First, the total number of tasks (200) is modest, reflecting a deliberate trade-off for the high-density, expert-intensive annotation required for reasoning evaluation. Second, the distribution of conditions is biased toward diseases with pronounced imaging correlates (e.g., stroke, tumors), with less representation of disorders that are primarily functional or metabolic. Third, although cases are drawn from multiple sources, the benchmark does not yet systematically capture the full spectrum of domain shift across hospitals, scanners, or reporting conventions. Fourth, the benchmark focuses on depth-oriented reasoning via multimodal and multi-field integration, leaving context-length robustness to complementary benchmarks. Finally, the current release represents only an initial stage of the benchmark, which will be progressively expanded over time.

These considerations underscore that `Neural-MedBench` is best interpreted as a high-resolution stress test for reasoning fidelity, not a comprehensive benchmark for statistical generalization. To address these limitations, future iterations will continuously grow in both scale and disease spectrum, with particular emphasis on underrepresented conditions, longitudinal follow-up, and multi-center variability, ensuring that the benchmark remains a continually evolving resource for the community.

## 7  CONCLUSION

In this work, we challenged the prevailing paradigm of evaluating medical VLMs through the lens of classification accuracy. We argued that this approach creates an "evaluation illusion," and proposed a more complete Two-Axis Framework that mandates complementary assessment of both Breadth (statistical generalization) and Depth (reasoning fidelity). To operationalize this missing Depth axis, we introduced Neural-MedBench, a compact, diagnostically complex benchmark for clinical neurology. Our extensive experiments yielded a sobering conclusion: a model's success on large-scale, Breadth-Axis benchmarks is a poor predictor of its ability to perform deep clinical reasoning. We found that even state-of-the-art models are consistently outperformed by human clinicians and that their failures stem primarily from fundamental breakdowns in reasoning, not perception.

While `Neural-MedBench` provides a crucial new perspective, we acknowledge its limitations. Its intentionally compact scale restricts broad statistical claims, and its current focus is on diagnostic reasoning rather than the full clinical workflow. The sample size for our human baseline, while offering vital insights, is also modest. These limitations, however, illuminate a clear and exciting roadmap for future work. The next generation of Depth-Axis benchmarks must expand in both scale and clinical scope, incorporating more diverse and underrepresented conditions, longitudinal data, and multi-center variability to systematically study domain shift. Notably, `Neural-MedBench` is designed as a continually evolving resource: the present set of cases and tasks reflects the state at submission, and its scale and coverage will be continuously extended.

Ultimately, `Neural-MedBench` is more than a dataset; it is a diagnostic tool for the field itself, providing a reproducible methodology to probe the core reasoning capabilities of our most advanced models. By offering this high-resolution lens and committing to its ongoing evolution, we aim to shift the community's focus from a narrow pursuit of accuracy towards the development of AI systems that possess genuine, trustworthy clinical reasoning. This benchmark serves as a foundational step in that critical journey.

## 8 ETHICS DETAILS

Data used in this study were obtained from the Alzheimer's Disease Neuroimaging Initiative (ADNI) database (adni.loni.usc.edu), the OASIS dataset, Radiopaedia, and case reports from articles published in The Journal of the American Medical Association (JAMA). All data were accessed with prior approval and are fully de-identified.

- **ADNI:** Investigators within ADNI contributed to data collection but did not participate in the analysis or writing of this manuscript. A comprehensive list of ADNI investigators is available at: ADNI Acknowledgement List. Data collection and sharing for ADNI were funded by the National Institute on Aging (U01 AG024904), the Department of Defense (W81XWH-12-2-0012), and other sponsors listed on the ADNI website. This manuscript was reviewed by the ADNI Data and Publications Committee prior to submission.
- **OASIS:** A portion of the OASIS dataset was restructured to comply with privacy protection protocols, ensuring that no personally identifiable information was included. Throughout the process, we adhered strictly to OASIS's privacy guidelines, safeguarding participant confidentiality.
- **Radiopaedia:** Radiopaedia data used in this study are provided under the Creative Commons Attribution-NonCommercial-ShareAlike 3.0 (CC-BY-NC-SA 3.0) license. The data were utilized for non-commercial purposes with proper attribution to Radiopaedia and are redistributed under the same licensing terms.

## 9 REPRODUCIBILITY STATEMENT

All experiments in this study are based on our `Neural-MedBench` dataset. Its sources, filtering, and construction process are described in Appendix A. The full dataset is publicly available on Hugging Face at https://huggingface.co/datasets/Reisen301/Neural-MedBench for reuse and verification.

## 10 ACKNOWLEDGMENTS

This work was supported in part by the Government Special Support Funds for the Guangdong Institute of Intelligence Science and Technology; in part by the Beijing Renyixun Health Technology Co., Ltd.; in part by the Young Scientists Fund of the National Natural Science Foundation of China under Grant 62506084 (M. Xu); and in part by the Young Scientists Fund of the National Natural Science Foundation of China under Grant 32500997 (S. Li).

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

## A ADDITIONAL DATASET DETAILS

**Data Source Breakdown.** To ensure transparency and reproducibility, we provide a detailed overview of the composition of `Neural-MedBench`. As shown in Table 4, the benchmark draws on four carefully vetted sources spanning public neuroimaging cohorts, curated online repositories, and peer-reviewed clinical case reports. Together these sources yield a heterogeneous pool of conditions ranging from common neurodegenerative diseases to rare atypical presentations, enabling a broad coverage of diagnostic challenges.

Table 4: Proportional breakdown of all cases in Neural-MedBench by source.

| Source | Proportion of Cases (%) | Primary Conditions Covered |
|---|---|---|
| ADNI | 18.2 | Alzheimer's Disease, Mild Cognitive Impairment |
| OASIS | 9.1 | Alzheimer's Disease, Normal Aging |
| Radiopaedia | 45.5 | Stroke, CNS Infections, Autoimmune Encephalitis, Tumors |
| JAMA Neurology | 36.4 | Rare Diseases, Atypical Presentations |

**Case Curation Protocol.** As illustrated in Figure 5, the construction of `Neural-MedBench` followed a rigorous, four-stage, funnel-shaped pipeline to distill an initial pool of over 2,000 candidate cases into the final 120 high-density cases. The stages were:

1. **Pooling and Filtering:** We first pooled cases from our sources and filtered them to retain only those with sufficient multimodal completeness (e.g., clear imaging, patient history, and clinical notes).

2. **Expert Curation:** A panel of senior neurologists reviewed the filtered cases for clinical plausibility, diagnostic complexity, and educational value, selecting for scenarios that require deep reasoning.

3. **Annotation of Ground Truth:** For each selected case, the expert panel systematically annotated the ground truth, which includes not only the final diagnosis but also differential diagnoses, key lesion characteristics, and the full explanatory reasoning chain.

4. **Consensus Review and Challenge Validation:** Finally, all annotated cases underwent a consensus review to resolve any disagreements. In this stage, we also performed a challenge validation using baseline models to filter out diagnostically trivial cases and ensure every case in the final benchmark presents a meaningful reasoning challenge.

This multi-stage process ensures that each of the 120 cases is diagnostically rich, clinically authentic, and serves as a potent test of a model's reasoning capabilities.

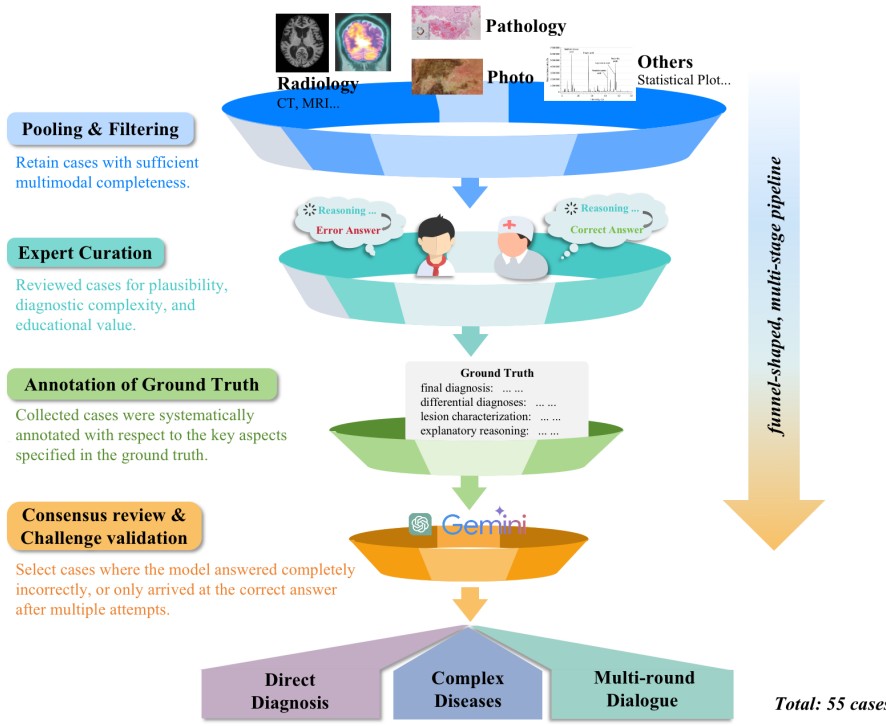

Figure 5: **The four-stage, funnel-shaped curation pipeline for `Neural-MedBench`.** The process distills over 2,000 initial candidates into 120 final cases by systematically filtering for multimodal completeness, curating for diagnostic complexity, annotating a rich ground truth, and validating the final reasoning challenge.

**Distribution of Clinical Conditions and Reasoning Tasks.** Figure 6 provides a dual view of the benchmark's diversity. The top panel (sunburst chart) illustrates the hierarchical distribution of the 120 clinical cases across a wide spectrum of neurological diseases. It demonstrates a balanced coverage of both common conditions like Alzheimer's disease and a long tail of rare or diagnostically challenging disorders (e.g., PERM, PAVF), ensuring a robust test of model generalization beyond frequency bias. The bottom panel details the distribution of the 200 questions across the three core cognitive capabilities probed by our benchmark: Differential Diagnosis (29%), Lesion Recognition (25%), and Rationale Generation (46%). The benchmark is thus heavily weighted towards higher-order reasoning, moving far beyond simple recognition.

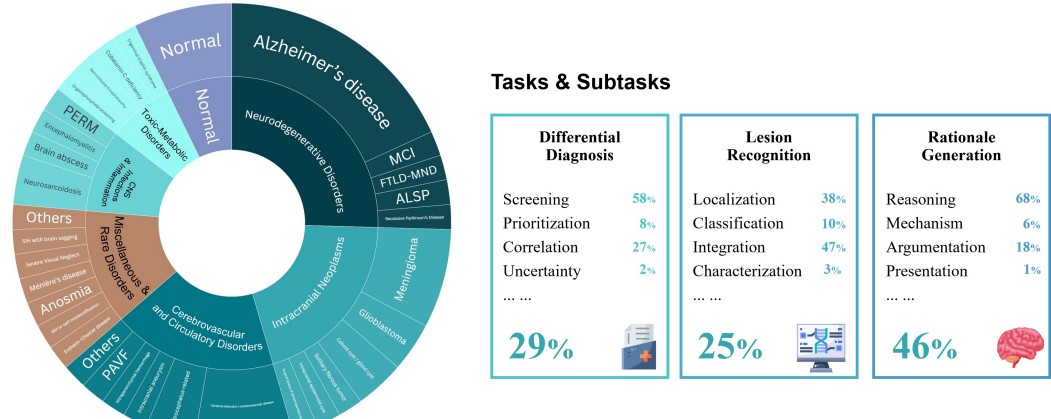

Figure 6: **Hierarchical Distribution of Neurological Diseases and Reasoning Tasks in `Neural-MedBench`.** (Top) The sunburst chart illustrates the distribution of the all clinical cases across a wide range of neurological conditions, from high-level categories (e.g., Neurodegenerative Disorders, Intracranial Neoplasms) to specific diagnoses (e.g., Alzheimer's disease, Glioblastoma), including a significant proportion of rare and complex disorders. (Bottom) The task distribution across the 200 questions, categorized into three core reasoning families: Differential Diagnosis (29%), Lesion Recognition (25%), and Rationale Generation (46%). Each family is further broken down into specific cognitive subtasks.

**Length Normalization.** We first segmented notes using standard clinical section headers (e.g., "History of Present Illness," "Neurological Examination," "Impression/Assessment," "Plan," etc.). We prioritized and always retained the core reasoning-relevant sections: HPI, Neuro Exam, and Impression/Assessment. Low-yield sections such as repeated "Past Medical History," exhaustive medication lists, or administrative details were truncated or removed if they exceeded a small threshold and were clearly redundant. We imposed an upper token budget per modality (e.g., per note) to avoid extreme outliers. When needed, we truncated from low-importance sections first, only truncating within high-priority sections as a last resort. Structured EHR fields (age, sex, key lab values) were not truncated. We did not pad or artificially inflate shorter notes. Normalization therefore acts mainly by trimming pathological "very long" tails, not forcing all samples to be identical in length. Natural variability in realistic note lengths (within a moderate range) is preserved.

**Content Statistics.** Figure 7 presents the detailed distributions of token counts and image sizes for the dataset. Panel (a) shows the distribution of English token counts in the clinical narratives, confirming that while we controlled for extreme length variations, a natural diversity remains. Panel (b) illustrates the variability in total image pixels per case, reflecting the real-world heterogeneity in MRI/CT acquisition protocols (e.g., number of sequences and slices). These statistics underscore that `Neural-MedBench` preserves authentic data characteristics while maintaining a balanced design for fair evaluation.

**Limitations of Dataset Composition.** While `Neural-MedBench` was deliberately curated for diagnostic depth, we acknowledge several limitations in its current composition. First, the total number of tasks (200) is modest, reflecting a deliberate trade-off for the high-density, expert-intensive annotation required for reasoning evaluation. Second, the distribution of conditions is biased toward diseases with pronounced imaging correlates (e.g., stroke, tumors), with less representation of disorders that are primarily functional or metabolic. Third, although cases are drawn from multiple sources, the benchmark does not yet systematically capture the full spectrum of domain shift across hospitals, scanners, or reporting conventions. Finally, the current release represents only an initial stage of the benchmark, which will be progressively expanded over time. These considerations underscore that `Neural-MedBench` is best interpreted as a high-resolution stress test for reasoning fidelity, not a comprehensive benchmark for statistical generalization. To address these limitations, future iterations will continuously grow in both scale and disease spectrum, with particular emphasis

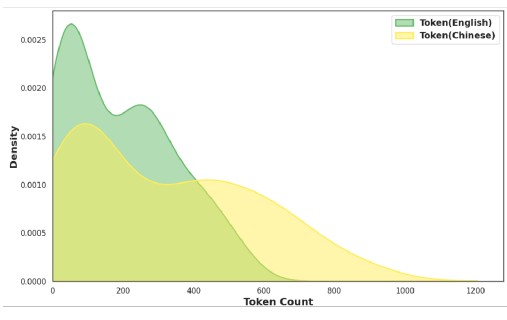
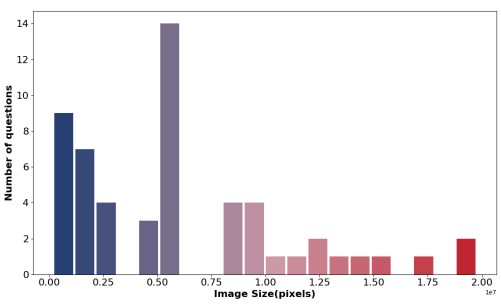

(a) Token Distribution Comparison        (b) Image Size Distribution

Figure 7: **Detailed content statistics for Neural-MedBench.**

on underrepresented conditions, longitudinal follow-up, and multi-center variability, ensuring that the benchmark remains a continually evolving resource for the community.

## B PROMPT TEMPLATES

We employed a structured role-prompting strategy to ensure that model behavior adhered to clinical logic.

> *"You are an experienced neurology expert, skilled in analyzing complex cases and providing preliminary diagnoses based on a combination of medical imaging and clinical descriptions. Please carefully answer the following questions based on the provided information."*

**Example Task Prompt.**

> **Patient history:** 47-year-old female with progressive motor weakness and intermittent visual disturbances.
> **Imaging:** Multi-sequence MRI (T2, FLAIR) uploaded below.
> **Task:** Please provide your most likely diagnosis, differential diagnoses, and explain your reasoning.

All multimodal inputs were interleaved with text. Imaging data were encoded in base64 and inserted as `<image>` tokens when supported by the model.

We employed a baseline model as an automatic judge to assess the agreement between the model's outputs and the reference answers. The evaluation is conducted using a fixed prompt.

> *You are an expert medical evaluator judging the accuracy of AI-generated diagnoses against reference answers.*
>
> *model_answer: [model_answer]*
> *ref_answer: [ref_answer]*
>
> *Guidelines:*
> *1. The model's diagnosis is correct if it captures the main pathology/condition mentioned in the reference diagnosis.*
> *2. Minor differences in terminology or additional details provided by the model are acceptable as long as the core diagnosis is correct.*
> *3. If the reference diagnosis contains multiple conditions, the model should identify at least the primary condition to be considered correct.*
> *4. Ignore differences in formatting, grammar, or level of detail if the core diagnosis is correct.*
>
> *Please provide a direct answer at the beginning, [yes] or [no].*

## C MODEL CONFIGURATIONS AND HYPERPARAMETERS

To ensure reproducibility, we list the exact model versions evaluated:

- GPT-4o: `gpt-4o-2025-03-26`
- GPT-5: `gpt-5-chat`
- Gemini 2.5-Pro: `gemini-2.5-pro-2025-06`
- Gemini 2.5-Flash: `gemini-2.5-flash-preview-04-17`
- Gemini 2.0-Flash: `gemini-2.0-flash-exp`
- Claude 4.0 Sonnet: `claude-sonnet-4-20250514`
- Claude 3.7 Sonnet: `claude-3-7-sonnet-20250219`
- Claude 3.5 Sonnet: `claude-3-5-sonnet-20241022`
- Qwen-VL-2.5: `qwen-vl-2.5-32b`
- Doubao: `doubao-1.5-vision-pro-32k-250115`
- LLaVA-Med: `llava-med-v1.5-mistral-7b`
- RadFM: `chaoyi-wu/RadFM`
- Med-Flamingo: `snap-stanford/med-flamingo`
- Huatuo-GPT-Vision: `Huatuo-gpt-vision-7b`
- MedGemma: `MedGemma-27B-it`
- Lingshu: `Lingshu-32B`

### C.1 HYPERPARAMETERS

Table 5 summarizes the key hyperparameter settings used across all experiments. For diagnostic accuracy (pass@1, pass@5), predictions were generated with nucleus sampling ($p = 0.95$, $T = 0.7$). For text similarity evaluation, we used BERTScore with `roberta-large`. All confidence intervals are reported using the Wilson score interval.

Table 5: Decoding hyperparameters used for each model

| Model | Hyperparameters |
|---|---|
| GPT-4o | temperature=0.7, top_p=0.95, seed=22 |
| GPT-5 | temperature=0.7, top_p=0.95 |
| Gemini 2.5-Pro | temperature=0.7, top_p=0.95 |
| Gemini 2.5-Flash | temperature=0.7, top_p=0.95 |
| Gemini 2.0-Flash | temperature=0.7, top_p=0.95 |
| Claude 4 Sonnet | temperature=0.7, top_p=0.95 |
| Claude 3.7 Sonnet | temperature=0.7, top_p=0.95 |
| Claude 3.5 Sonnet | temperature=0.7, top_p=0.95 |
| Qwen-VL-2.5 | temperature=0.7, top_p=0.95 |
| DouBao | temperature=0.7, top_p=0.95 |
| LLaVA-Med | temperature=0.7, top_p=0.95 |
| MedGemma | temperature=0.7, top_p=0.95 |
| RadFM | temperature=0.7, top_p=0.95 |
| Med-Flamingo | temperature=0.7, top_p=0.95 |
| HuatuoGPT | temperature=0.7, top_p=0.95 |
| Lingshu | temperature=0.7, top_p=0.95 |

## D    CONFIDENCE INTERVALS

For diagnostic accuracy metrics, we report 95% confidence intervals using the Wilson score interval:

$$CI = \frac{1}{1 + \frac{z^2}{n}} \left( \hat{p} + \frac{z^2}{2n} \pm z \sqrt{\frac{\hat{p}(1 - \hat{p})}{n} + \frac{z^2}{4n^2}} \right) \tag{1}$$

where $\hat{p}$ is observed accuracy, $n$ is sample size, and $z = 1.96$ for 95% confidence.

## E    GRADER BIAS CONSIDERATIONS

A potential concern arises from the use of GPT-4o both as a benchmarked model and as the primary LLM-based grader. To mitigate self-evaluation bias, we conducted a rigorous clinician-in-the-loop calibration, finding a very high correlation with expert consensus scores (Pearson's $r > 0.9$). In addition, our evaluation protocol is hybridized with semantic similarity metrics (BERTScore) and direct clinician validation, ensuring that grading robustness does not rely solely on GPT-4o. To ensure our LLM-grader is a reliable proxy for expert judgment, we conducted a rigorous validation. The key steps and findings are:

**1. Human Gold Standard:** A panel of three board-certified neurologists independently scored a large subset of 100 model responses to establish a human consensus score.

**2. Quantitative Correlation:** The crucial finding was a very high correlation (Pearson's r>0.9) between our LLM-grader's scores and the human consensus, scientifically validating its reliability.

**3. Bias Check:** We specifically checked for self-evaluation bias and found no systematic advantage for GPT-4o's own outputs compared to those from other models.

For future iterations, we plan to expand the grader pool by incorporating multiple LLMs (e.g., Claude, Gemini) and additional clinician review to further enhance fairness and reproducibility.

## F    ERROR TAXONOMY ANNOTATION PROTOCOL

To better understand failure modes, two board-certified neurologists independently annotated 100 randomly sampled incorrect responses using the following taxonomy:

- **Reasoning Failure:** Correctly observed features but incorrect causal inference.
- **Perceptual Failure:** Misinterpretation of visual features (e.g., lesion missed).
- **Visual Hallucination:** Fabricated findings not present in the input data.
- **Knowledge Gap:** Missing medical knowledge required for correct diagnosis.

Across the incorrectly answered cases, the distribution of primary error types was: Reasoning Failure 51% , Perceptual Failure 27%, Visual Hallucination 13%, and Knowledge Gap 9%. These proportions suggest that current multimodal language–vision systems for neuroimaging are limited more by inference and decision-making than by raw visual recognition or encyclopedic recall.

Disagreements were resolved by consensus. Inter-rater agreement was $\kappa = 0.82$, indicating strong consistency.

**Reasoning Failure (51%).** The dominant failure mode reflects cases in which salient image findings were noted but mapped to an incorrect diagnosis or pathophysiologic explanation. Typical patterns included misweighting across MRI sequences (e.g., over-interpreting FLAIR hyperintensity without integrating diffusion restriction), conflating acute and chronic stigmata, and collapsing multi-lesion presentations into a single etiology. These errors point to gaps in causal and temporal modeling—models can "see" but struggle to structure evidence, adjudicate competing hypotheses, or apply exclusion criteria.

**Perceptual Failure (27%).** Pure detection/characterization mistakes were the next largest category. Contributing factors included low signal-to-noise, motion artifact, and small lesion size.

**Visual Hallucination (13%).** In nearly one-fifth of failures, the model asserted findings that were not present (e.g., "midline shift," "ring-enhancing lesion") and then built a diagnosis on those fictitious observations. This is a distinct safety risk because it couples perceptual fabrication with persuasive language.

**Knowledge Gap (9%).** Pure deficits in medical knowledge were least common. Notably, the relatively small share here underscores that, for common entities, the models "know enough" but still reason unreliably.

## G  ERROR CASES

To provide a more granular and intuitive understanding of the failure modes quantified in our main analysis, this section presents a series of representative error cases from `Neural-MedBench`. Figure 8 presents a Level 2 complex diagnosis task. This task specifically tests the model's ability to link a pharmacological history to specific neuroimaging findings, a common real-world clinical challenge. The subsequent pages display 14 additional failure cases, presented in groups of figure (Figure 9 to Figure 12). Together, these qualitative examples provide concrete evidence for the limitations of current VLMs and highlight the critical importance of Depth-Axis evaluation.

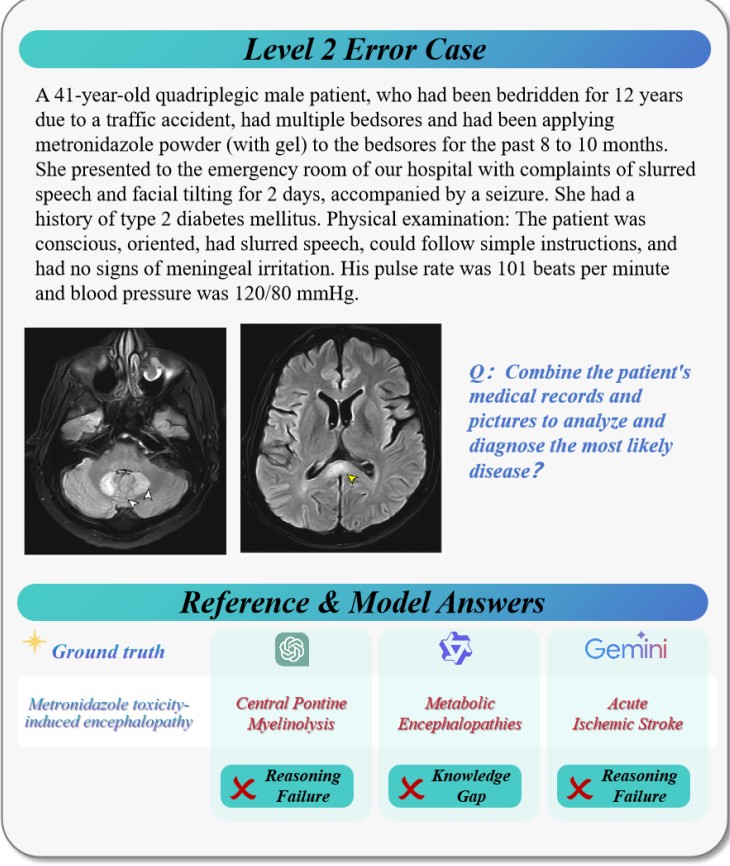

Figure 8: **A representative Level 2 failure case demonstrating both Reasoning Failure and Knowledge Gap.**

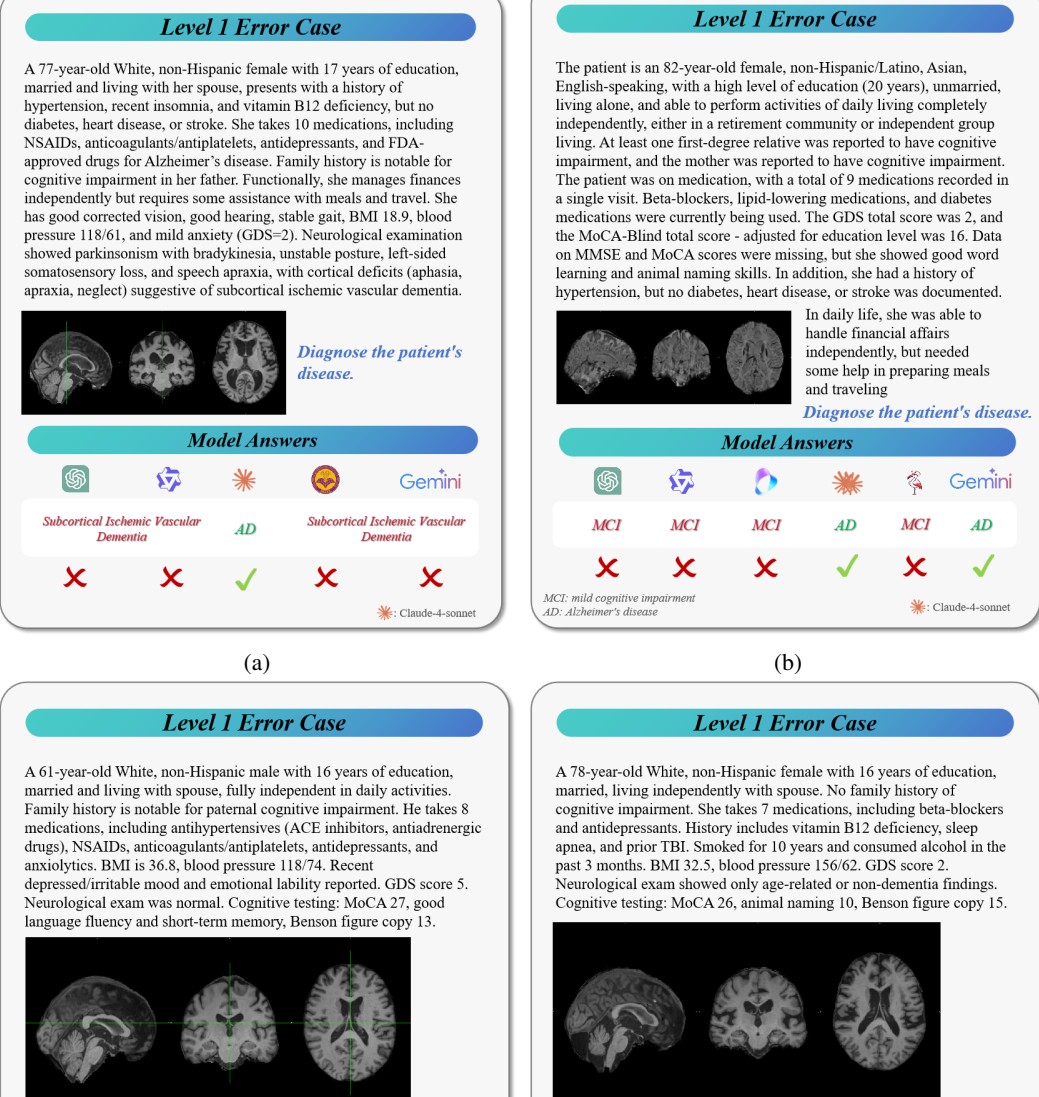

Figure 9: **Additional error cases.**

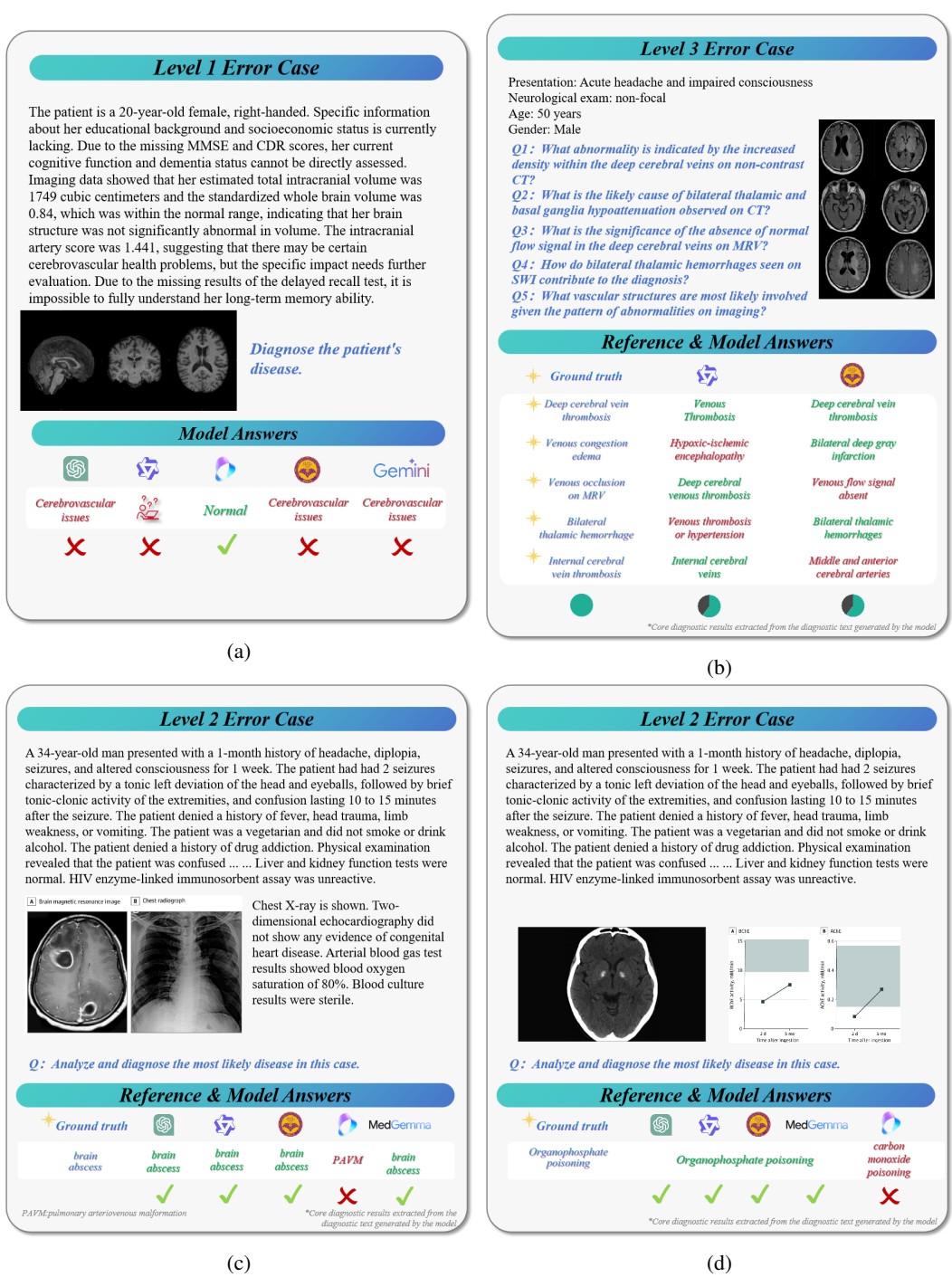

Figure 10: **Additional error cases.**

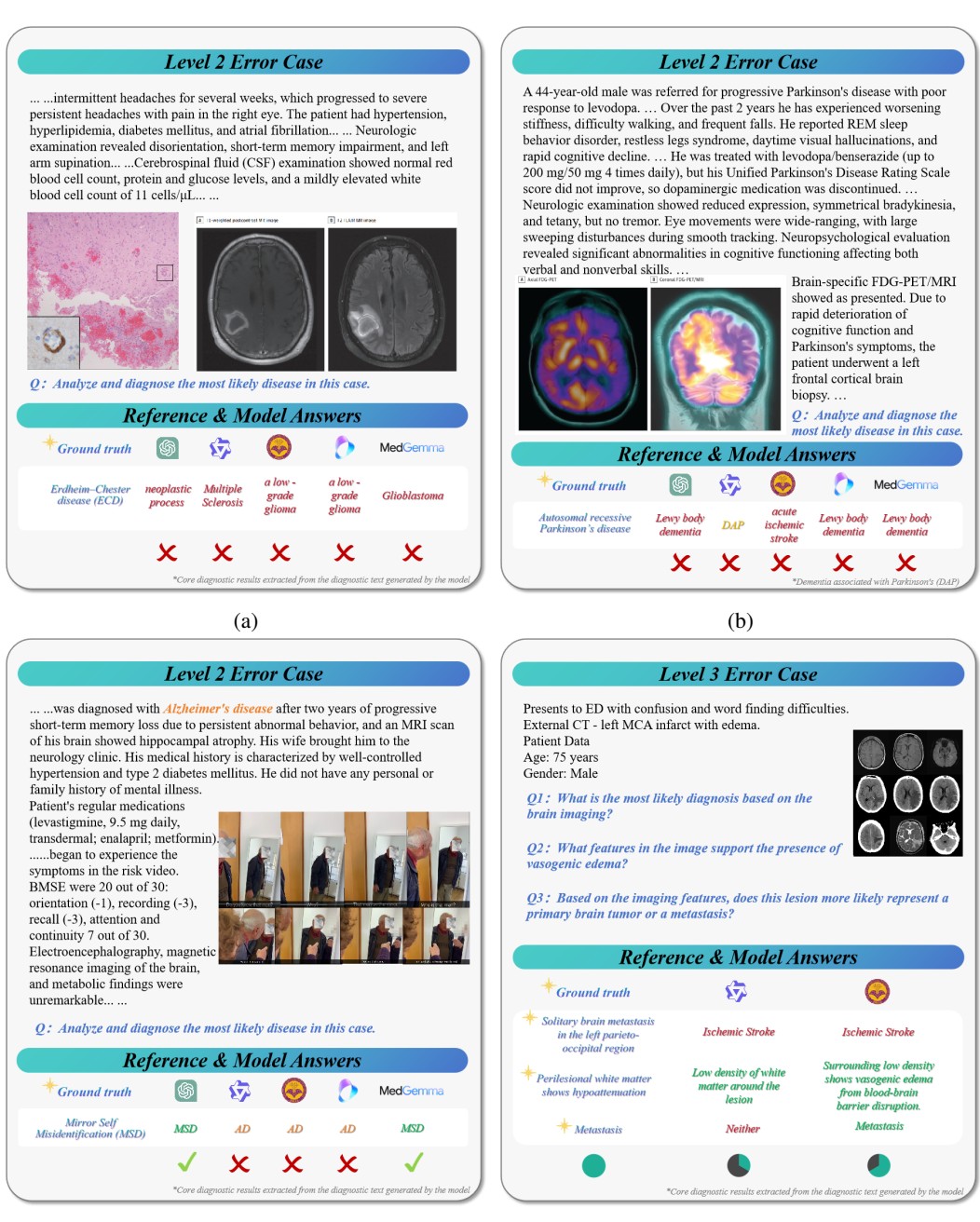

Figure 11: **Additional error cases.**

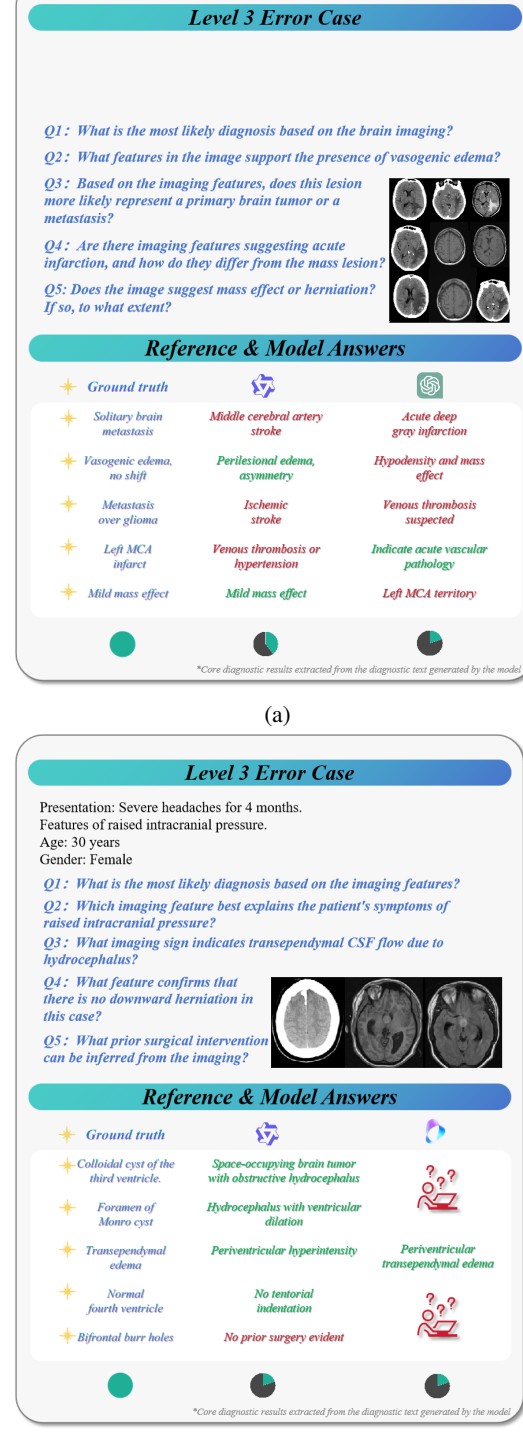

(a)

(b)

Figure 12: **Additional error cases.**

