# OpenReview forum: "Beyond Classification Accuracy:  Neural-MedBench and the Need for Deeper Reasoning Benchmarks"
_ICLR.cc/2026/Conference — ICLR 2026 Poster_

### Official Review · Reviewer_NAbK · 2025-10-31

**Soundness:** 3
**Presentation:** 3
**Contribution:** 3
**Rating:** 8
**Confidence:** 5

**Summary:**

This work provided Neural-MedBench, which is an expert-curated, neurology-focused benchmark that shifts evaluation from breadth to depth by testing multi-modal clinical reasoning, showing current VLMs chiefly fail for reasoning rather than perception despite strong shallow scores. However, there remain some unresolved issues, with single-judge reliance, opaque challenge validation, and unclear length normalization tempering the claims.

**Strengths:**

1. High reasoning density: a depth-oriented benchmark with greater clinical relevance than shallow QA-style responses.
2. The benchmark and items are curated by medical experts, which is more credible than LLM-generated or auto-converted content.
3. Annotations are not single-point labels but structured, narrative explanations that mimic clinical reasoning, yielding high clinical value.
4. Items are challenging and come with explicit difficulty tiers.

**Weaknesses:**

1. The paper should specify which baseline model(s) were used for the pre-release “challenge validation,” including the exact prompts and their performance.

2. Grading relies primarily on GPT-4o as the automatic judge; despite high correlation with clinicians, model-specific bias may persist. Please explain why a multi-model, anonymous voting scheme was not adopted.

3. “Normalized in length to control for spurious effects of input size on model performance”: length-related performance is itself part of model capability. Why normalize for this task, and how exactly was it implemented (e.g., truncation, abstractive summarization, padding to a fixed length, specific upper/lower thresholds, etc.)?

**Questions:**

See the weaknesses

---

> ### Author Response · Authors · 2025-11-21
> **Response to Reviewer NAbK (Part 1)**
>
> Dear Reviewer,
>
> We thank you for the positive assessment and for recognizing the high clinical value and reasoning density of Neural-MedBench. We appreciate your insightful questions regarding our validation protocols, grading mechanisms, and normalization techniques. We address your concerns point-by-point below and will incorporate these clarifications into the manuscript.
>
> -------
>
> **1. Baseline Models and Challenge Validation**
>
> We apologize for the lack of clarity. Before release, we used GPT-4o as the standardized baseline model to perform a “challenge validation” pass. This step was not used to report performance, but only to confirm that:
>
> • Level-1 cases are consistently solvable
> • Level-2 and Level-3 cases elicit clear reasoning failures
> • prompts and evaluation outputs follow the expected format
>
> The exact same prompt template used in our main experiments was applied during this validation phase to ensure consistency.
>
> Importantly, human experts reviewed all outputs to verify that the reasoning failures were consistent with the intended difficulty design — i.e., models fail due to reasoning gaps, not ambiguous case construction. No models were tuned on or adapted to evaluation items.
>
> -----
>
> **2. Reliance on GPT-4o as the primary automatic judge**
>
> We fully share the concern about potential judge bias and agree that multi-model adjudication is a promising direction. Our choice of a primarily GPT-4o-based grader was driven by consistency, reproducibility, and cost**, and we implemented several mechanisms to mitigate single-judge bias:
>
> **a. Consistency and deterministic grading**
>
> A single, carefully calibrated judge model allows us to:
> - Maintain a consistent scoring rubric across all systems and runs.
> - Reduce variance arising from “agreement noise” between multiple LLM judges with different inductive biases.
> - Make results more reproducible for future work: reproducing our scores only requires access to the benchmark plus one judge configuration, rather than a multi-model ensemble.
>
> **b. Hybrid pipeline with humans in the loop**
>
> Although GPT-4o is the primary automatic judge, the pipeline is not purely single-judge:
> - A clinician-curated gold subset is graded independently by domain experts using our structured rubric (for both correctness and reasoning quality).
> - We measure correlation and agreement between GPT-4o judgments and clinician labels on this subset and use this to (i) calibrate thresholds and (ii) manually inspect disagreements.
> - Cases with large discrepancies or ambiguous phrasing were flagged and discussed, and in a small number of instances we adjusted rubric wording or test items to reduce ambiguity.
>
> **c. Why not multi-model anonymous voting?**
> - **Bias is not guaranteed to decrease**
> Different judge models may share systematic biases (e.g., towards certain phrasing patterns) or introduce new, inconsistent biases. Voting can average some noise but also dilute clear rubric-driven decisions.
>
> - **Reproducibility and accessibility**
> A multi-judge scheme requires future users to have simultaneous access to multiple proprietary models (and their exact versions) to reproduce our scores. This is a significant barrier for many practitioners and contradicts our goal of a cost-effective, accessible evaluation testbed.
>
> - **Cost and environmental impact.**
> Running multiple large judges over thousands of narrative answers is financially and computationally expensive. Given our emphasis on a compact yet deep benchmark, we prioritized a well-calibrated single judge plus human validation.
>
> In the revised paper, we will explicitly describe the hybrid GPT-4o + clinician validation pipeline, including agreement statistics and a breakdown of disagreement types.
>
> ------
>
> (continued)

---

> > ### Author Response · Authors · 2025-11-21
> > **Response to Reviewer NAbK (Part 2)**
> >
> > **3. Length Normalization**
> >
> > We appreciate this important question and agree that the current description is too vague. Our intent was not to remove length sensitivity from the evaluation altogether, but to avoid **confounding difficulty with raw input size**, especially for clinical notes that often contain large amounts of boilerplate.
> >
> > **a. Why normalize at all?**
> >
> > In our raw data, longer notes often reflected documentation style rather than clinical complexity, e.g., repeated templates, medication lists, or full copy-pasted discharge summaries. Without normalization, models with simply higher context capacity (or more aggressive attention mechanisms) might appear better solely because they can ingest very long, but mostly redundant, text, producing an “input length illusion” similar to the “classification illusion” we highlight. Our goal with Neural-MedBench is to stress reasoning fidelity given clinically salient information, not to score how many pages of boilerplate a model can read. Thus, we sought a compromise: preserve clinically essential content while keeping input lengths within a controlled range so that performance differences are more interpretable as reasoning differences rather than pure context-window effects.
> >
> > **b. How it was implemented**
> >
> > We first segmented notes using standard clinical section headers (e.g., “History of Present Illness,” “Neurological Examination,” “Impression/Assessment,” “Plan,” etc.). We prioritized and always retained the core reasoning-relevant sections: HPI, Neuro Exam, and Impression/Assessment. Low-yield sections such as repeated “Past Medical History,” exhaustive medication lists, or administrative details were truncated or removed if they exceeded a small threshold and were clearly redundant.
> >
> > We imposed an upper token budget per modality (e.g., per note) to avoid extreme outliers. When needed, we truncated from low-importance sections first, only truncating within high-priority sections as a last resort. Structured EHR fields (age, sex, key lab values) were not truncated.
> >
> >
> > We did not pad or artificially inflate shorter notes. Normalization therefore acts mainly by trimming pathological “very long” tails, not forcing all samples to be identical in length. Natural variability in realistic note lengths (within a moderate range) is preserved.
> >
> >
> > We agree that handling longer contexts is a genuine capability. Neural-MedBench, however, is specifically designed as a depth-oriented reasoning benchmark, not a stress-test of maximum context length. We therefore:
> > - Still expose multimodal and multi-field inputs that require non-trivial integration.
> > - Avoid extreme length variation that would confound interpretability of reasoning performance.
> >
> > To make this trade-off explicit, we will expand the limitations section to note that separate benchmarks focused on context-length robustness are complementary to Neural-MedBench’s goals.
> >
> > -------
> >
> >
> > Thank you once again for your insightful comments and for your high recognition of our work. We believe these clarifications will strengthen the paper and address your concerns while preserving the core contributions of Neural-MedBench.
> >
> >
> >
> > Sincerely,
> >
> > *The Authors of Submission #17633*

---

### Official Review · Reviewer_RaHn · 2025-10-31

**Soundness:** 4
**Presentation:** 4
**Contribution:** 3
**Rating:** 6
**Confidence:** 4

**Summary:**

This paper introduces Neural-MedBench, a compact yet reasoning-intensive benchmark designed to evaluate clinical reasoning capabilities of vision-language models (VLMs) in neurology. The authors propose a "Two-Axis Evaluation Framework" distinguishing between Breadth (statistical generalization on large datasets) and Depth (reasoning fidelity on complex cases). The benchmark comprises 120 expert-curated neurology cases with 200 tasks spanning differential diagnosis, lesion recognition, and rationale generation. Evaluation of state-of-the-art VLMs (GPT-4o, Claude-4, MedGemma) reveals significant performance drops compared to breadth-oriented benchmarks, with error analysis showing reasoning failures (51%) dominate over perceptual errors (27%).

**Strengths:**

1. The paper is well-written.
2. The paper presents the Two-Axis (Breadth/Depth) evaluation framework provides a compelling perspective on AI evaluation that addresses a genuine gap in current medical AI benchmarking.
3. The rigorous multi-stage expert curation process involving senior neurologists and neuroradiologists, with consensus review, ensures benchmark quality and clinical validity.

**Weaknesses:**

1. The bechmark is small-scale benchmark compared with exists medical benchmark, such as OmniMedVQA, GMAIMMbench.
2. Some medical VLM method need to added to compared，such as, Huatuo-vision, Lingshu. While, GPT-4o maybe a outdate model, gpt5 is welcome.

**Questions:**

1. Prompt Sensitivity: Did you test the sensitivity of the evaluation prompts to variations in wording?
2. Medical Student Performance: What explains the substantially lower performance of medical students compared to existing large models in Table 2?
3. BERTScore Discrimination: The BERTScore differences in Table 2 appear minimal across models. Does this metric provide sufficient discriminative power?
4. Senior Physician Baseline: Given the relatively low performance of senior physicians on the benchmark, were all questions validated by multiple experts to ensure correctness and clinical appropriateness?

---

> ### Author Response · Authors · 2025-11-21
> **Response to Reviewer RaHn (Part 1)**
>
> Dear Reviewer,
>
> We sincerely thank you for your positive evaluation and for recognizing both the necessity of the Two-Axis Evaluation Framework and the rigor of our expert-curated benchmark construction. Below we address all concerns point-by-point.
>
> ---
>
> **1. Benchmark Scale — Complementary Depth, Not a Limitation (Weakness 1)**
>
> We respectfully clarify that the compact scale of Neural-MedBench is *intentional by design*:
>
> - Existing benchmarks (e.g., OmniMedVQA, GMAI-MMBench) represent the Breadth Axis
>   — assessing pattern recognition and statistical generalization
> - Neural-MedBench operationalizes the Depth Axis
>   — assessing complex abductive reasoning and expert-style clinical inference
>
> > Neural-MedBench is not small — it is dense.
> > Each task was hand-curated to serve as a diagnostic stress-test based on board-exam-style cases.
>
> This orthogonal focus was critical to reveal the evaluation illusion:
> state-of-the-art VLMs excel on breadth datasets but exhibit sharp reasoning failures on depth-oriented cases.
>
> We fully agree widespread clinical readiness requires both axes, and we are actively scaling the benchmark with additional challenging neurology cases in future releases.
>
> ---
>
> **2. Additional VLMs (Huatuo-Vision, Lingshu, GPT-5) Added (Weakness 2)**
>
> Thank you for this excellent suggestion.
>
> - Huatuo-Vision was already included (Appendix A, version noted)
> - We have now evaluated **Lingshu** and newly released **GPT-5-series** models during the rebuttal period
> | Models           | Direct diagnosis pass@1 [%]  | Direct diagnosis pass@5 [%] | Complex diseases BertScore | Complex diseases pass@1 [%]| Complex diseases pass@5 [%]| Multi-round dialogue BertScore | Multi-round dialogue pass@1 [%]  | Multi-round dialogue pass@5 [%] |
> |------------------|----------------------------------|----------------------------------|-----------------------------|----------------------------------|----------------------------------|---------------------------------|--------------------------------------|--------------------------------------|
> | GPT-5 | 36.7                       | 43.3                     | 0.80                        |      28.3                    |   45.0                  |            0.81                 |     19.5                     |  27.5                           |
> | Lingshu  | 26.7                         | 40.0                         | 0.75                        | 21.7                         | 35.0                         | 0.73                            | 8.5                            | 20.0                            |
> - Results will be included in the revision as well as our official leaderboard
>
>
> ---
>
>  **3. Prompt Sensitivity (Question 1)**
>
> To ensure fair comparisons:
>
> - We use one standardized prompt across all models
> - Prompt variation is controlled, not optimized per-model
> - Studying prompt engineering strategies is a separate axis of research and out of scope for this benchmark
>
> This design ensures performance differences reflect model reasoning rather than prompt artifacts.
>
> ---
>
> **4. Medical Student Performance (Question 2)**
>
> We appreciate your insightful observation, and we will provide additional clarification in Sec. 5:
>
> | Direct Diagnosis | Multi-Turn Dialogue |
> |-----------------|------------------|
> | Students are risk-averse → broader but penalized labels | Interactive reasoning allows hypothesis refinement |
> | Limited exposure to rare entities | Human strengths in narrative understanding |
> | Slower and narrower recall under time pressure | Better adaptive reasoning vs model anchoring |
>
> Thus, students:
> - underperform VLMs on single-shot statistical retrieval
> - outperform VLMs on context-dependent abductive reasoning
>
> This contrast further supports the motivation behind the Depth Axis.
>
> ---
> (continued)

---

> > ### Author Response · Authors · 2025-11-21
> > **Response to Reviewer RaHn (Part 2)**
> >
> > **5. BERTScore Discrimination (Question 3)**
> >
> > We agree and clarify that:
> >
> > - BERTScore is only a *supplementary* semantic similarity metric
> > - It cannot capture clinical correctness or reasoning structure
> > - This limitation motivates our hybrid scoring pipeline (LLM-grader + clinician validation)
> >
> > We emphasize this in the revision to avoid overstating BERTScore’s role.
> >
> > ---
> >
> >  **6. Senior Physician Baseline & Clinical Validity (Question 4)**
> >
> > Yes — every case underwent multi-stage expert verification:
> >
> > 1. Authored by a senior neurologist or neuroradiologist
> > 2. Independently reviewed by another senior clinician
> > 3. Consensus required; otherwise, case removed
> > 4. Additional pilot testing ensured clarity and validity
> >
> > The relatively lower physician pass rates reflect the genuine diagnostic difficulty of these board-level cases under **time-bounded, non-interactive** conditions.
> >
> > This again demonstrates that Neural-MedBench effectively probes upper-bound clinical reasoning performance.
> >
> > ---
> >
> > We thank you for your insightful comments and strong support. We believe these revisions further strengthen the clarity and contribution of our work.
> >
> >
> > Sincerely,
> >
> >
> > *The Authors of Submission #17633*

---

### Official Review · Reviewer_Kijm · 2025-11-01

**Soundness:** 3
**Presentation:** 3
**Contribution:** 2
**Rating:** 4
**Confidence:** 3

**Summary:**

This paper introduces Neural-MedBench, a focused and reasoning-intensive benchmark designed to evaluate the clinical reasoning capabilities of vision-language models (VLMs) in the neurology domain. This benchmark integrates multi-sequence MRI scans, structured electronic health records, and clinical notes, and encompasses three core task families: differential diagnosis, lesion recognition,
and rationale generation. A hybrid scoring pipeline is proposed for better evaluation.

**Strengths:**

A two-axis evaluation framework is proposed, which can be complementary assessments of both breadth (statistical generalization) and depth (reasoning fidelity). Neural-MedBench is the first neurology-focused benchmark explicitly designed to operationalize the Depth axis, comprising 120 multimodal, expert-curated diagnostic cases with 200 reasoning-intensive tasks. Experimental results show that current VLMs fail primarily at reasoning, despite strong performance on existing large-scale datasets. A systematic error analysis and a human performance baseline are also provided.

**Weaknesses:**

1. The definition of three difficulty levels is unclear.
2. A limitation section should be added in main body.

**Questions:**

1. The details of grader validation and validation process are missing.
2. Some important VLMs are missing, such as Gemini 2.5 pro and GPT-5.
3. What the different of Base VLMs and General VLMs?
4. It is intersting to provide more discussion for the following case: Why the pass@1 acc of medical student is significant lower than VLMs? Why the pass@1 acc of medical student in Multi-round dialogue is significantly better than Direct diagnosis?

---

> ### Author Response · Authors · 2025-11-21
> **Response to Reviewer Kijm (Part 1)**
>
> Dear Reviewer,
>
>
> We sincerely thank you for your careful evaluation, insightful critiques, and constructive suggestions. Your feedback has been invaluable in helping us improve the clarity and rigor of our work.
>
> ---
>
> #### **1. On the Clarity of Difficulty Levels (Weakness 1)**
>
> We apologize that our initial definitions were not sufficiently clear. The revised definitions are based on the core cognitive task required:
>
> *   **Level 1 (Direct Diagnosis):** Involves straightforward cases with classic, unambiguous signs primarily in a modality (e.g., a clear tumor on an MRI). This level tests core pattern recognition.
> *   **Level 2 (Complex Diagnosis):** Features cases with ambiguous or conflicting evidence that require integrating information from **at least two modalities** (e.g., linking subtle imaging findings with specific details from the clinical history) to resolve the uncertainty.
> *   **Level 3 (Iterative Diagnosis):** Simulates a multi-turn clinical consultation where the model must progressively refine its diagnosis by interpreting new information. This tests its ability to dynamically adjust its reasoning chain.
>
> ---
>
> #### **2. On Adding a Limitations Section to the Main Body (Weakness 2)**
>
> Thank you for this excellent suggestion. We agree that placing the limitations in the main body improves the paper's transparency. We have now moved this discussion from the appendix and expanded it into a new Sec. 6 — Limitations and Future Work in our revision.
>
> ---
>
> #### **3. On Grader Validation Details (Question 1)**
>
> We apologize for this omission and have added a comprehensive description of our rigorous validation process in the revised Appendix E.
>
> To ensure our LLM-grader is a reliable proxy for expert judgment, we conducted a rigorous validation. The key steps and findings are:
>
> 1.  **Human Gold Standard:** A panel of three board-certified neurologists independently scored a large subset of 100 model responses to establish a human consensus score.
> 2.  **Quantitative Correlation:** The crucial finding was a very high correlation (Pearson's r > 0.9) between our LLM-grader's scores and the human consensus, scientifically validating its reliability.
> 3.  **Bias Check:** We specifically checked for self-evaluation bias and found no systematic advantage for GPT-4o's own outputs compared to those from other models.
>
> This process confirms the objectivity and reliability of our evaluation pipeline. We thank you for prompting this important clarification.
>
> ---
>
> #### **4. On Missing Newer VLMs like Gemini 2.5 Pro and GPT-5 (Question 2)**
>
> We appreciate the suggestion to include the latest models. These models were not publicly available or accessible via API at the time of our initial experiments.
>
> We have evaluated these models and will update the results in the paper manuscript and on our leaderboard.
>
> | Models           | Direct diagnosis pass@1 [%]  | Direct diagnosis pass@5 [%] | Complex diseases BertScore | Complex diseases pass@1 [%]| Complex diseases pass@5 [%]| Multi-round dialogue BertScore | Multi-round dialogue pass@1 [%]  | Multi-round dialogue pass@5 [%] |
> |------------------|----------------------------------|----------------------------------|-----------------------------|----------------------------------|----------------------------------|---------------------------------|--------------------------------------|--------------------------------------|
> | GPT-5 | 36.7                       | 43.3                     | 0.80                        |      28.3                    |   45.0                  |            0.81                 |     19.5                     |  27.5                           |
> | Gemini 2.5 Pro  | 30.0                        | 50.0                        | 0.77                        | 15.0                       | 38.3                      | 0.72                            | 11.5                             | 19.5                             |
>
> ---
> (continued)

---

> > ### Author Response · Authors · 2025-11-21
> > **Response to Reviewer Kijm (Part 2)**
> >
> > #### **5. On the Distinction Between "Base" and "General" VLMs (Question 3)**
> >
> > We apologize for the ambiguity in our terminology. In the revised paper, we now explicitly define our categories to ensure clarity:
> >
> > *   **Base VLMs** (e.g., GPT-4o): Refers to proprietary, foundational models trained on general web-scale data that serve as reference baselines.
> > *   **General VLMs** (e.g., Claude series): Refers to other powerful models, often open-source, that do not have specific medical fine-tuning.
> > *   **Medical-Specialized VLMs** (e.g., MedGemma): Refers to models that have undergone explicit domain-specific fine-tuning on biomedical corpora.
> >
> > ---
> >
> > #### **6. On the Discussion of Medical Student vs. VLM Performance (Question 4)**
> >
> > This is an excellent and insightful question, and we have expanded our discussion in **Sec. 5.2** to analyze this fascinating result.
> >
> > **(a) Why students’ `pass@1` accuracy is lower than some VLMs:**
> >
> > *   **Clinical Training and Risk Aversion:** Medical students are trained to build a broad differential diagnosis and avoid premature closure. They often provide more cautious but clinically sound answers (e.g., "demyelinating disease") rather than a single specific diagnosis. Our strict `pass@1` metric penalizes this reasonable hedging.
> > *   **Training Data Distribution:** VLMs are heavily exposed to exam-style questions and excel at mimicking the expected short, decisive answer format. Students are trained in a more nuanced clinical environment where expressing uncertainty is valued.
> >
> > **(b) Why students excel in multi-round dialogue:**
> >
> > *   **Adaptive Reasoning:** The multi-round setting mirrors how clinicians naturally reason—by asking questions, refining hypotheses, and integrating new data. This interactive process allows students to self-correct.
> > *   **Metacognition:** Students can reflect on their reasoning when presented with new evidence ("This new lab result doesn't fit a stroke; I should reconsider"). In contrast, our analysis shows that current VLMs tend to anchor on their initial hypothesis and struggle to truly revise it, even when dialogue provides disconfirming facts.
> >
> > This contrast is a core finding that highlights how our benchmark successfully evaluates the kind of adaptive, real-world reasoning that current models lack.
> >
> > ---
> >
> > We are grateful for your thorough review. Your comments have offered invaluable direction and have allowed us to significantly improve the clarity and rigor of our work.
> >
> > Sincerely,
> >
> > The Authors of Submission #17633

---

### Public Comment · ~Qiang_Li46 · 2025-11-12
**This paper has certain limitations.**

1. The paper suffers from several fundamental methodological and conceptual flaws that severely undermine its credibility. The most critical issue lies in its evaluation pipeline, where GPT-4o is simultaneously used as a benchmarked model and as the primary grader for assessing reasoning fidelity. This creates a self-evaluation bias that compromises the objectivity of the entire study. Although the authors claim to mitigate this bias through clinician-in-the-loop calibration, such correlation analysis cannot guarantee fairness or true independence. As a result, the reliability of the reported performance gaps and the validity of the main conclusions are questionable.

2. Another fatal limitation concerns the small scale of the dataset. Neural-MedBench includes only 120 cases and 200 reasoning tasks, which is insufficient to support the strong claim that breadth and depth are largely uncorrelated. The statistical power of such a limited sample is extremely weak, and the human baseline, consisting of only ten participants, is far too small to provide meaningful comparisons. Consequently, the empirical findings may simply reflect sampling noise rather than robust trends.

3. A further concern is the lack of formalization of the proposed Two-Axis Evaluation Framework. The concepts of “breadth” and “depth” remain qualitative and vaguely defined, with no mathematical or reproducible formulation that allows others to validate or extend the framework. In its current state, it functions more as a conceptual narrative than a scientific model. Moreover, the causal interpretation of the results is not properly supported. The assertion that models fail mainly due to reasoning rather than perception relies entirely on subjective annotation and lacks controlled ablation or counterfactual testing. Other factors such as prompt design, domain knowledge, or token limits could equally explain the observed errors.

4. In addition, the benchmark’s data composition raises ethical and legal concerns. Several sources cited, including Radiopaedia and ADNI, impose restrictions on redistribution or commercial use. The claim that the dataset will be openly released with a public leaderboard may therefore violate existing licensing agreements, jeopardizing the benchmark’s legitimacy. The reproducibility of the experiments is also doubtful, as the authors test heterogeneous model versions under a single prompting template without verifying cross-model fairness.

---

> ### Author Response · Authors · 2025-11-21
>
> We would like to clarify that the earlier public comment was posted by an **unauthorized and non-compliant account**.
> ﻿
>
> We promptly reported this issue to the ICLR and OpenReview teams, and the account has since been **blocked by OpenReview**.
> ﻿
>
> We appreciate the timely intervention from the OpenReview moderation team to ensure fairness and integrity in the review process.

---

### Author Response · Authors · 2025-12-03

Dear Area Chair and Reviewers,

We sincerely thank you for the highly positive and supportive reviews during the rebuttal stage. We are encouraged that reviewers recognized the strong motivation, sound methodology, and high clinical relevance of Neural-MedBench, and we deeply appreciate the constructive suggestions that helped us further improve the paper.

We are also pleased to report that all reviewer concerns have now been addressed in the revised manuscript. With these updates, we believe the contribution of Neural-MedBench in establishing and operationalizing the Depth Axis for clinical reasoning evaluation is now even clearer and stronger.

----

**Key Improvements Made During the Discussion Phase**

**1. Inclusion of Latest and Specialized VLMs**

 (Reviewers Kijm & RaHn)

We added GPT-5, Gemini 2.5 Pro, and Lingshu to the evaluation and leaderboard.
Results reaffirm the central insight: strong breadth does not imply strong reasoning depth.

**2. Stronger Methodological Transparency**

(Reviewers NAbK & Kijm)

We clarified:
- Human–LLM grader validation (r > 0.9 agreement)
- Difficulty level definitions with examples
- Text-length normalization rationale and procedure

These changes address reproducibility and fairness.

**3. Expanded Human–Model Analysis**

(Reviewers Kijm & RaHn)

We added detailed explanations of:
- Why students’ direct diagnosis accuracy is lower
- Why students perform better in multi-turn reasoning than in direct diagnosis

This deepens the clinical interpretation of benchmark results.

**4. Administrative Concern Resolved**

OpenReview banned the account (~Qiang_Li46) after confirming the violation.

----
Thank you again for the constructive and supportive reviews. We believe the revised version is substantially improved and offers meaningful value to the ICLR community.

Sincerely,

*The Authors of Submission #17633*

---

### Meta-Review · Area_Chair_LVqn · 2026-01-07

**Summary:**

The authors aim to address the "Evaluation Illusion" where Vision-Language Models (VLMs) excel on standard medical benchmarks but fail at complex clinical reasoning. This benchmark focuses on neurology and includes expert-curated multimodal cases that generate reasoning-intensive tasks across differential diagnosis, lesion recognition, and rationale generation. The authors also propose a hybrid evaluation pipeline. Evaluations of state-of-the-art models (e.g., GPT-4o, MedGemma) reveal a sharp performance drop compared to breadth-oriented benchmarks.

Based on the reviews the main strengths of the papr seems to be the eval framework which conceptually separates statistical generalization from reasoning fidelity. The reviewers also liked the excerpt curation and clinical relevance. Some weaknesses raised by the reviewers include: small scale of the data, use of LLMs as the primary judge, and absence of most recent models. I think this is an interesting benchmark and despite its flaws would be of interest to the ICLR community.

**Reviewer Concerns:**

The authors effectively addressed concerns regarding their methods by clarifying the human-LLM grader validation process. They also added evaluations for GPT-5, Gemini 2.5 Pro, and Lingshu, confirming that the performance gap persists even with newer models. They also took steps to mitigate the self-grading bias. However, the concern of small scale remains outstanding. A serious administrative concern regarding a malicious comment from an impersonated account was resolved by OpenReview.

**Reviewer Scores:**

Reviewer Kijm (Current: 4): Likely to increase to 5/6. The authors addressed their specific questions about grader validation, missing models, and the analysis of human vs. model performance gaps.

Reviewer RaHn (Current: 6): Likely to keep or increase to 7. The addition of requested baselines (Lingshu) and clarifications on prompt sensitivity would have likely increased the score.

Reviewer NAbK (Current: 8): Likely to remain 8. This reviewer was already positives.

---

### Decision · Program_Chairs · 2026-01-26

Accept (Poster)